# Epigenetic inheritance is unfaithful at intermediately methylated CpG sites

Amir D. Hay [1], Noah J. Kessler [1], Daniel Gebert[1], Nozomi Takahashi[1], Hugo Tavares [1], Felipe K. Teixeira [1,2] ✉ & Anne C. Ferguson-Smith [1] ✉

DNA methylation at the CpG dinucleotide is considered a stable epigenetic mark due to its presumed long-term inheritance through clonal expansion. Here, we perform high-throughput bisulfite sequencing on clonally derived somatic cell lines to quantitatively measure methylation inheritance at the nucleotide level. We find that although DNA methylation is generally faithfully maintained at hypo- and hypermethylated sites, this is not the case at intermediately methylated CpGs. Low fidelity intermediate methylation is interspersed throughout the genome and within genes with no or low transcriptional activity, and is not coordinately maintained between neighbouring sites. We determine that the probabilistic changes that occur at intermediately methylated sites are likely due to DNMT1 rather than DNMT3A/3B activity. The observed lack of clonal inheritance at intermediately methylated sites challenges the current epigenetic inheritance model and has direct implications for both the functional relevance and general interpretability of DNA methylation as a stable epigenetic mark.

The establishment and maintenance of global DNA methylation patterns are essential for the development and function of vertebrate genomes[1,2]. Despite being known as a stable epigenetic mark[3–6], accumulating evidence indicates that at a given CpG dinucleotide, DNA methylation status may vary between cell divisions, suggesting that DNA methylation patterns are more dynamic than previously anticipated. One mechanism that can explain this is the dynamic binding of factors and chromatin states that modulate methylation deposition during development[7]. In contrast, some early investigations found that intermediately methylated sites could spontaneously arise within subclonal cell populations derived from single cells[8,9]. Other groups observed that intermediate methylation is inconsistently inherited after cell divisions and therefore represents either error in maintenance or spontaneous de novo methylation in a range of developmental and tumour cell populations[10–12].

Conceptually, in each cell, there are only three possible symmetric methylation states that can occur at a single CpG site: unmethylated (0%), methylated on one allele (50%), or methylated on both alleles (100%). Therefore, it is thought that intermediate levels of DNA methylation (10–90%) in somatic cells represent population averages of these symmetric and faithfully inherited methylation states[13–15]. While models of how intermediate methylation may be subsequently inherited through cellular divisions have been proposed[16,17], the extent of such dynamics has not been systematically examined at a genome-wide scale, nor have the principles dictating DNA methylation stability through clonal propagation been determined.

## Results

### Evaluating the fidelity of DNA methylation inheritance through clonal expansion

We devised an experiment to assess the fidelity of DNA methylation through clonal expansion at the genome-scale, by subcloning populations of cells and performing target DNA capture followed by high-throughput bisulfite sequencing (tcBS-seq, see the "Methods" section) on both the subclone and parent populations of cells (Fig. 1A). To do so, we established mouse embryonic fibroblasts (MEFs) from two sibling E13.5 mouse embryos and immortalised the cell lines. From these parental lines, we randomly sampled 14 single cells to establish

[1]Department of Genetics, University of Cambridge, Downing Street, Cambridge CB2 3EH, UK. [2]Department of Physiology, Development and Neuroscience, University of Cambridge, Downing Street, Cambridge CB2 3DY, UK. ✉e-mail: fk319@cam.ac.uk; afsmith@gen.cam.ac.uk

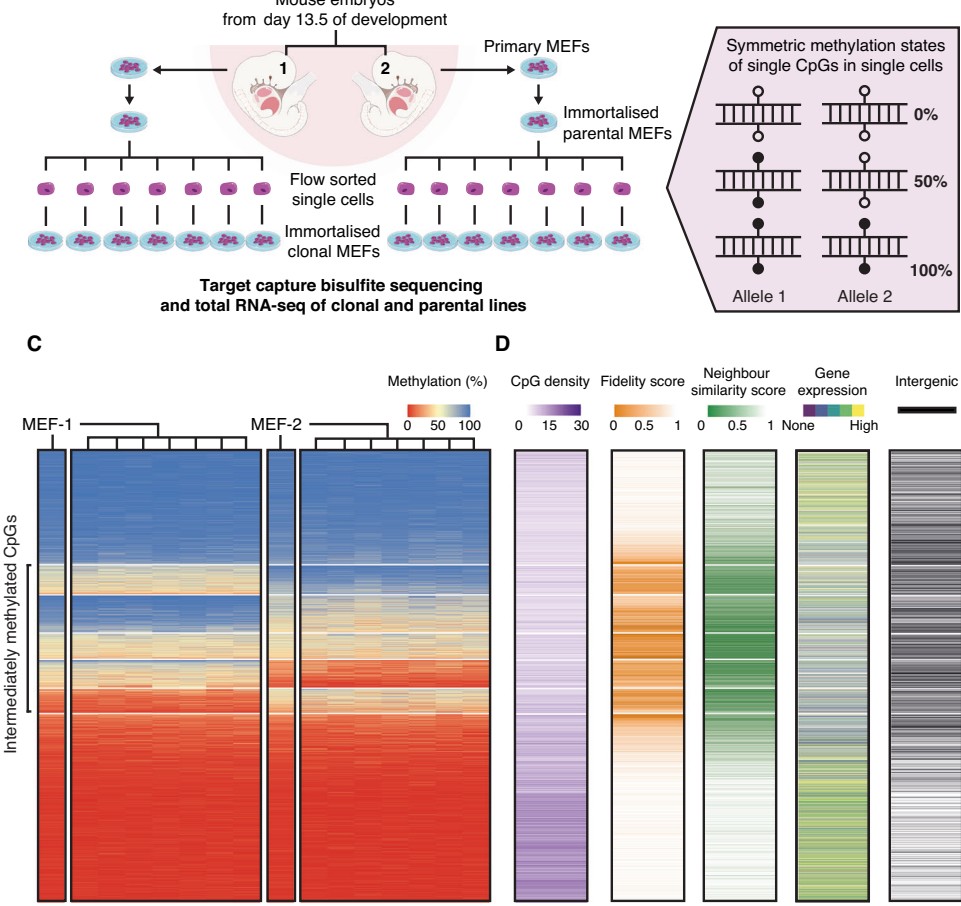

**Fig. 1 | Intermediately methylated sites show lower epigenetic inheritance fidelity. A** Mouse embryonic fibroblasts (MEFs) were isolated from E13.5 mouse embryos and immortalised. The resulting cell lines, MEF-1 and MEF-2, are referred to as the "parental" lines. Single cells were randomly selected by flow cytometry from these lines and grown into derivative "clonal" lines. Target capture bisulfite sequencing and total RNA sequencing were performed on both the parental and clonal lines. **B** Illustration showing that at the single-cell level, there are only three strand-symmetric methylation states that can exist at a single CpG dinucleotide: 0%, 50%, and 100%. **C** Heatmaps of 1,203,687 CpGs sorted by median methylation (%) within *k*-means clusters ($k = 7$, separated by white lines). The two parental lines

are denoted as MEF-1 and MEF-2, with the seven clonal lines shown to the right of the corresponding parental line. **D** Heatmaps of CpG density (in purple) calculated as the number of CpGs within 100 bp of each focal CpG, methylation fidelity score (in orange) calculated as a proxy for the retention of symmetric methylation states from the single cell to a clonal line population, neighbour similarity score (in green) as an approximation of the consistency of methylation between neighbouring CpGs in a clonal line, and transcription quintiles derived from gene expression averages across all the cell lines (no expression in purple, high expression in yellow). Intergenic CpGs (blank lines) are characterised by a lack of overlap with an annotated protein-coding transcript or promoter.

clonal populations of around 1–2 million cells on which we profiled DNA methylation using tcBS-seq. In total, the methylation levels of >1.2 million CpGs (or around 5% of CpGs in the mouse genome, with a median coverage of 32x per CpG per dataset; Supplementary Fig. 1) were determined across 16 samples and used for downstream analysis. Of note, CpGs covered by tcBS-seq are enriched for early-replicating genic regions of the genome and are underrepresented at transposable elements (Supplementary Tables 1 and 2).

In principle, in a single cell at a single CpG site, there are three possible stable (strand-symmetric) methylation states (Fig. 1B), which provides us with a tractable framework to determine the fidelity of DNA methylation inheritance through clonal expansion. In a purely faithful scenario, the DNA methylation status of each CpG in the clonal lines should be exclusively either 0%, 50%, or 100%.

Globally, methylation data generated from parental and clonal lines revealed that the methylation state of most CpGs (66%) is consistent across both parental lines and all 14 clonal samples (Fig. 1C, Supplementary Figs. 1 and 2). When CpGs are classified according to their methylation states using *k*-means clustering (Supplementary Fig. 3A and B), we find that 40% of all tested CpGs are consistently

hypomethylated (U) across all analysed lines and are enriched for CpG dense regions (Fig. 1D and Supplementary Fig. 3C). Conversely, 26% of CpGs are consistently hypermethylated (M). As expected, the frequency of these two states indicates that the majority are explained by faithful inheritance of DNA methylation. In agreement, CpGs that are consistently hypo/hypermethylated in MEF-1-derived lines, exhibit the same state in MEF-2-derived lines. On the other hand, 33% of CpGs are shown to be intermediately methylated across all clonal lines derived from at least one of the parental cell lines (Fig. 1C). Among these intermediately methylated CpGs, most (26% of all analysed CpGs) are consistently hypo/hypermethylated across one set of lines but intermediately methylated in the other set (UI or MI, respectively). In addition, a subset of CpGs (7%) was intermediately methylated across all the cell lines (I).

To determine the degree of faithful methylation inheritance at the CpG level, we calculated the fraction of clonal lines that exhibit 0%, 50% or 100% methylation. We refer to this metric as the fidelity score, based on the concept that faithful transmission of the methylation state would result in only 0%, 50%, or 100% methylation in the clonal line populations derived from single cells. As expected from the

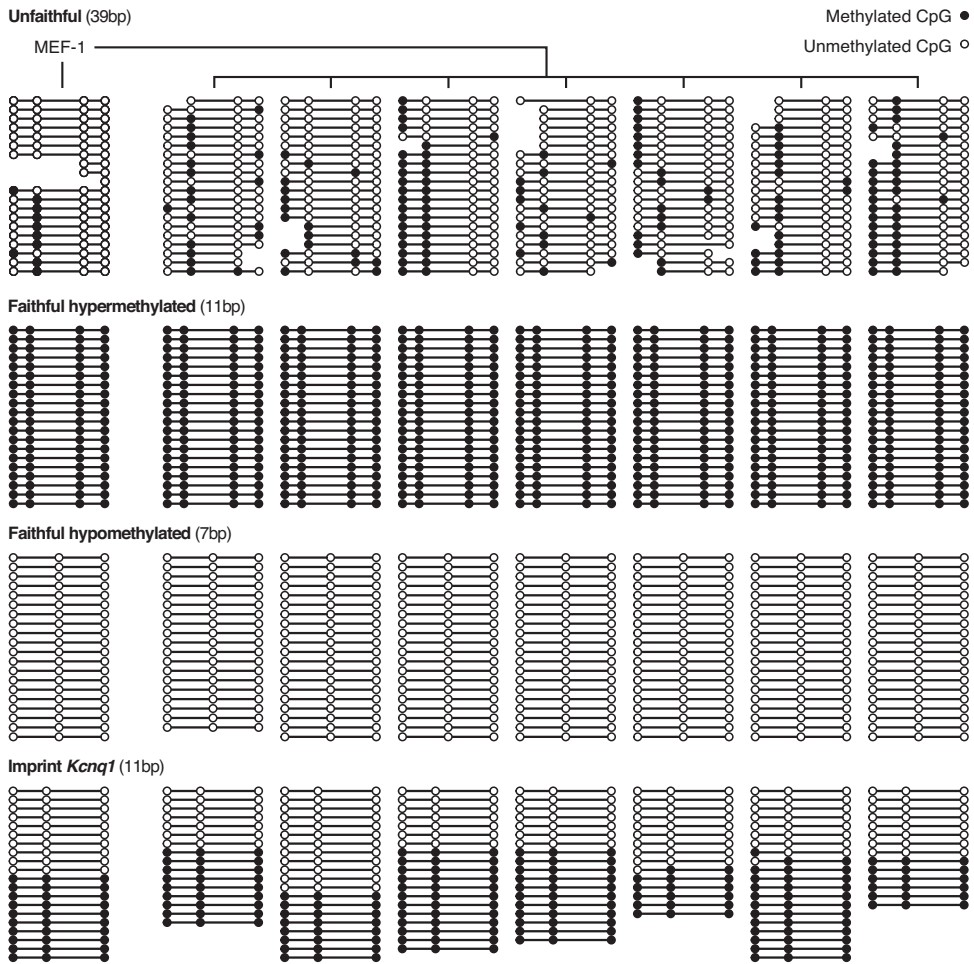

**Fig. 2 | Examples of unfaithful, faithful and imprinted regions of the genome.** Unfaithful regions show a sporadic methylation pattern that contrasts with the consistency of methylation states at faithful and imprinted regions of the genome. The parental line is denoted as MEF-1, with the seven clonal lines shown to the right. Reads from the tcBS-seq data are represented by lines and CpGs are shown as circles with accurate mapping of respective distance in base pairs between them. Methylated CpGs are shown as filled circles, while unmethylated CpGs are shown as open circles. Reads are ordered by average methylation of the covered CpG sites. The example regions shown above are each represented by between 15 and 20 reads at the following genomic coordinates: (mm10): unfaithful (chr15:79,251,875–79,251,914), faithful hypermethylated (chr1:34,302,911–34,302,922), faithfully hypomethylated (chr4:124,751,396–124,751,403) and imprint Kcnq1 (chr7:143,295,599–143, 295,610).

consistency of methylation states across the samples, we find that CpGs classified as M and U generally have a high fidelity score. On the other hand, CpGs that have the potential to be intermediately methylated have a significantly lower fidelity score (Fig. 1D and Supplementary Fig. 3D). Therefore, intermediate methylation states generally represent the unfaithful inheritance of methylation, and are characterised by the inconsistency of methylation patterns between parental and clonal lines (Fig. 2 and Supplementary Figs. 4–19).

**The relationship between methylation fidelity and transcription**
To gain insight into the principles dictating DNA methylation stability, with the basis that intermediate levels reflect unfaithful inheritance through clonal expansion, we compared the methylation groups with respect to their genomic sequence context. Because CpG density is a major predictor of methylation status[18], we first asked how CpG density correlates with the different methylation groups. As expected for consistently hypomethylated CpGs[19], we found that U CpGs reside in CpG-dense regions in comparison to the other groups. On the other hand, M CpGs, as well as CpGs that have potential to be intermediately methylated (UI, I, MI), have significantly lower CpG density (Fig. 1D and Supplementary Fig. 3C). Given that methylation can be spatially regulated across multiple neighbouring CpGs[20], we calculated a

methylation co-variation score between each one of the 1.2M CpGs and its closest neighbour (neighbour similarity score, see the "Methods" section). We found that, compared to CpGs classified as U and M, CpGs with potential for intermediate methylation (UI, I, MI) are less likely to have a similar methylation level to the closest CpG (Fig. 1D and Supplementary Fig. 3E). Together, these results show that intermediately methylated CpGs, which are generally unfaithfully propagated, are enriched in regions of low CpG density, and attain methylation independently of neighbouring CpGs.

Since DNA methylation is associated with transcriptional repression[21], we performed total RNA-sequencing on the parental and clonal MEF lines to investigate whether there is a relationship between the fidelity of methylation and gene expression. First, we classified all protein-coding genes (21,835) into five expression level groups ranging from "none" (~30% of genes) to "high" (~20% of genes) using the average normalised expression values from MEF-1 and MEF-2 RNA-seq datasets (Supplementary Fig. 20). We observed that intermediately methylated CpGs (UI, I, and MI) are more likely to be located within promoters and bodies of genes that are not expressed, or expressed at low levels, whereas U and M CpGs are more likely to be located within highly expressed genes (Fig. 1D and Supplementary Fig. 21, A and B). Moreover, we found that CpGs classified as intermediately methylated

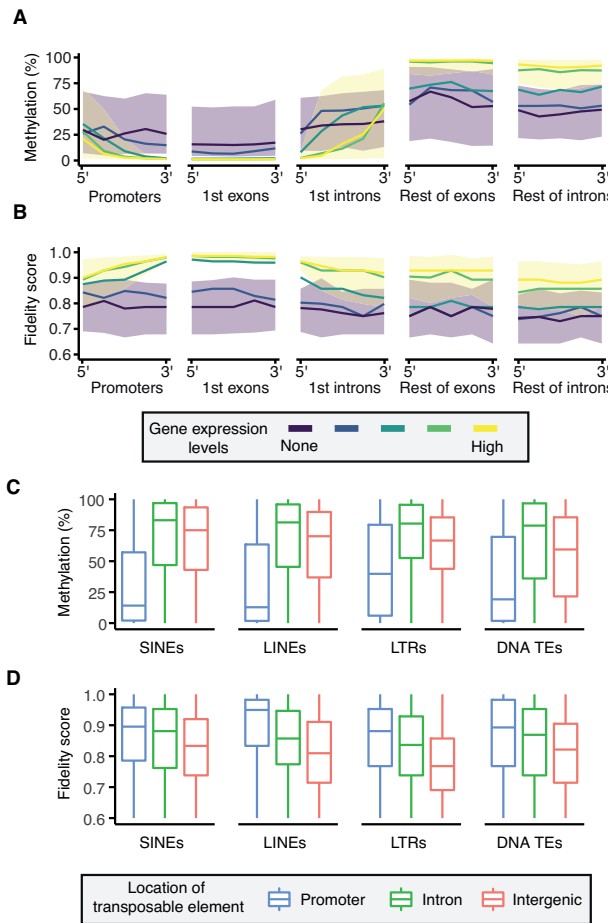

**Fig. 3 | Unfaithful intermediate methylation associates with the lack of transcription. A** Methylation levels and **B** fidelity score characterised along the following regions of protein-coding genes: promoters, 1st exons, 1st introns, the rest of exons, and the rest of introns. Each region is split into five tiles at which median (lines) or interquartile range (ribbons) of methylation or fidelity score is shown. Only regions covered by at least three CpGs are considered; single-exon genes are excluded. Gene expression levels are represented by colours ranging from purple (no expression) to yellow (high expression). Interquartile range ribbons for both methylation and fidelity score are only shown for the genes that are not expressed (light purple) or highly expressed (light yellow). **C** Boxplots showing methylation and **D** fidelity score of transposable elements (SINEs [$n = 2145$], LINEs [$n = 597$], LTRs [$n = 2816$], and DNA transposons [$n = 338$]) that exist in promoter (blue; # of SINEs = 316, # of LINEs = 58, # of LTRs = 105, # of DNA TEs = 35), intronic (green; # of SINEs = 926, # of LINEs = 246, # of LTRs = 611, # of DNA TEs = 128), or intergenic regions (orange; # of SINEs = 903, # of LINEs = 293, # of LTRs = 2100, # of DNA TEs = 175) of the genome. Methylation levels and fidelity score are calculated as the mean across CpGs in each TE. For **C** and **D**, the box of the boxplot shows the 25th, 50th, and 75th percentiles; the whiskers extend to 1.5*IQR beyond the edges of the box (where IQR = 75th−25th percentile), with outliers shown as dots.

(UI, I, and MI) are more likely to be intergenic compared to U and M CpGs (Fig. 1D and Supplementary Fig. 21C). These results reveal a relationship between transcriptional activity and the stability of DNA methylation across the samples.

Methylation levels differ between promoters, which are typically hypomethylated, and gene bodies, which are frequently hypermethylated[22]. Consistent with this, at highly expressed genes, we found that DNA is hypomethylated at the promoter and first exon, and hypermethylated at the rest of the exons and introns (Fig. 3A, Supplementary Fig. 22A, and Supplementary Table 4). The methylation dynamics of highly expressed genes are matched with a high fidelity

score throughout the entire gene (Fig. 3B and Supplementary Fig. 22B). This pattern is in sharp contrast to what is observed in genes with no or low expression. In this case, the entire genic region, including promoters and gene bodies alike, is enriched for intermediately methylated CpGs with low methylation fidelity. This suggests that either transcriptional activity defines the patterns of faithful hypo- and hypermethylation throughout a gene, or that a faithful methylation state contributes to expression. Furthermore, it indicates that the absence of transcription may be permissive for the presence of unfaithful intermediate methylation levels.

Transposable elements (TEs) are potentially mobile genetic units that can be transcriptionally repressed by DNA methylation[23,24]. However, we found that many CpGs within TEs are intermediately methylated (Fig. 3C). Therefore, we asked whether intermediate methylation associates with either the age and/or has a particular distribution within TE sequences. For example, recently integrated TEs are more likely to retain transcriptional potential and therefore may be preferentially targeted by DNA methylation[24]. We assessed the relationship between methylation status of a CpG and the evolutionary age of the TE it overlaps (where age is measured as the percent divergence of the individual element from the TE consensus sequence) (Supplementary Fig. 23). Globally, TE sequence divergence does not correlate with DNA methylation level nor fidelity (Supplementary Fig. 23B and C). Intermediate methylation of CpGs within TEs is thus unlikely to be related to the age of the element. Moreover, we could not find an association between intermediate methylation levels and the position of a CpG within a TE. Using SINEs as a tractable model, we observed that methylation levels are similar irrespective of the CpG position along the element (Supplementary Fig. 23D), even though like genes, SINEs are structured and have their own promoter regions[25]. Taken together, these results indicate that intermediate CpG methylation within TEs is not dependent on TE sequence divergence and is equally distributed along TE sequences, at least in SINEs.

Besides being frequently found in intergenic regions, TEs can also exist in genic regions such as promoters and introns (Supplementary Fig. 24), but rarely in exons[26,27]. Given that promoters and gene bodies generally have distinct patterns of methylation, we tested whether intermediate methylation levels at TEs can be determined by the genomic features of the TE insertion site. We observed that TEs in promoters are generally hypomethylated and TEs in introns tend to be hypermethylated (Fig. 3C). TEs in intergenic regions tend to be less methylated than those in introns, but higher than those in promoters. Indeed, intermediate methylation of CpGs within TEs is more likely to occur within intergenic regions, which is consistent with our observation that CpGs in intergenic TEs have lower fidelity scores compared to those in promoters and introns (Fig. 3D). Therefore, the lack of transcriptional activity at the insertion locus is strongly associated with intermediate methylation within TEs, and in turn its fidelity.

**Intermediate methylation states are probabilistically heritable**

The fidelity score allowed us to determine that intermediate methylation states are unfaithfully propagated through clonal expansion. However, this metric does not include the parental line methylation state, and therefore cannot be used to determine how states may be transmitted between the parental and clonal cell lines. Conceptually, we propose two ways by which DNA methylation at a given CpG is transmitted through cellular divisions: faithful and stochastic (Fig. 4A). A faithful process would result in the clonal lines being enriched for 0%, 50%, and 100% methylation states in proportions that recapitulate the parental methylation state. Whereas a stochastic process would result in the clonal lines being enriched for a distribution centred around the parental average methylation level. With low (0–10%) or high (90–100%) parental methylation states, the clonal lines tend to recapitulate the parental methylation level, better fitting a faithful process (Fig. 4B and Supplementary Fig. 25A). However, for the "low

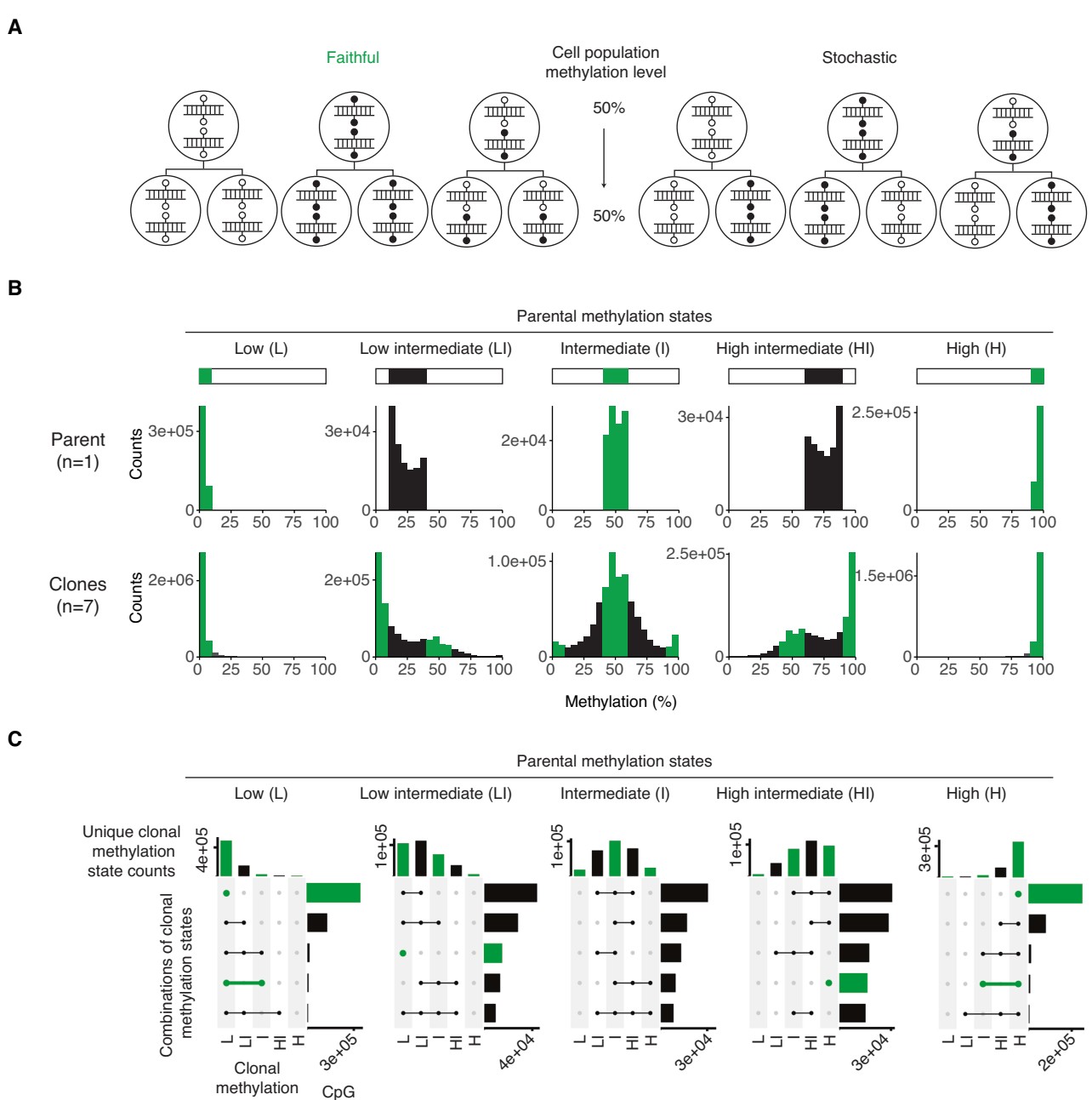

**Fig. 4 | Intermediately methylated CpGs are prone to probabilistic inheritance through clonal expansion. A** Two ways by which DNA methylation at a given CpG site can be transmitted through clonal expansion: faithful and stochastic. Each large circle represents a cell—the top row represents the parental cells, and the bottom row represents the daughter cells that arise from cell division. Symmetric methylation at a single CpG site is illustrated as a small filled-in circle on either one or both alleles, whereas the absence of methylation is illustrated as a small empty circle. **B** Clonal line methylation distributions from different parental line methylation states for the MEF-1 cell lines. **C** UpSet plots of clonal line methylation states per CpG from different parental line methylation states for the MEF-1 cell lines. In each panel, the top bar plots show the number of unique clonal methylation states represented per CpG. The horizontal bar plots show the CpG counts that exhibit a certain combination of clonal methylation states. Only the five most representative clonal methylation state combinations are shown. Green bars represent cases of potential faithful methylation inheritance because this kind of methylation inheritance will only result in 0%, 50%, or 100% methylation states in the clonal lines. Similarly, green dots and lines in the UpSet plots represent cases and combinations of potential faithful methylation inheritance. Methylation states are defined quantitatively as the following: Low = 0–10%, Low intermediate = 10–40%, Intermediate = 40–60%, High intermediate = 60–90%, High = 90–100% methylation.

intermediate" (10–40%), "intermediate" (40–60%), and "high intermediate" (60–90%) parental methylation states, the clonal lines display modal and skewed distributions that are not reflective of strictly faithful methylation inheritance.

To better visualise these clonal methylation dynamics, we used UpSet plots of the clonal methylation data split by the parental methylation state (Fig. 4C and Supplementary Fig. 25B). With respect to an initial parental methylation state, the top bar plots depict the frequency of distinct states observed amongst the clonal lines for a given CpG. For any given parental state, the most frequently observed state per clonal line CpG is the same as the parental one. The side plots reveal the most common combinations of states observed amongst

the derived lines. Unsurprisingly, low (0–10%) and high (90–100%) parental methylation states result most commonly in the same low or high methylation states in the clonal lines, respectively. However, any intermediate parental methylation state (10–40%, 40–60%, 60–90%) most commonly results in a combination of two or three states in the clonal lines, which includes the original state. Hence the heritability of an intermediate state is neither purely faithful nor stochastic. Rather, this suggests a probabilistic inheritance of intermediate methylation states, which allows for some, but not perfect, heritability of the cell population methylation level between the parental and clonal lines.

To determine what epigenomic features could be associated with the differential inheritance of methylation states, we calculated fold enrichment for various histone tail modifications that overlap with clonal or probabilistically methylated CpGs (Supplementary Fig. 26A, B and Supplementary Table 5; see the "Methods" section). We find that histone tail modifications that are generally associated with transcriptional repression (H3K27me3 and H3K9me3[28,29]) are enriched at the probabilistic intermediately methylated CpGs (Supplementary Fig. 26C). Faithfully methylated CpGs are enriched for histone tail modifications that are associated with transcriptional activation at both hypomethylated promoters (H3K27ac, H3K9ac, H3K4me3[30–32]) and hypermethylated gene bodies (H3K36me3[33]). These findings are in line with our finding noted above that unfaithful methylation is associated with intergenic regions and unexpressed genes (Fig. 3).

### De novo methyltransferases Dnmt3a/3b are not required for perpetuating intermediate methylation states

The existence of probabilistic unfaithful methylation inheritance suggests that there is continuous loss and gain of methylation at intermediately methylated CpG sites. To test whether DNMT3A/3B is mechanistically responsible for the probabilistic acquisition of methylation, we generated four MEF lines from *Dnmt3a^flox/flox^3b^flox/flox* mice and used recombinant TAT-CRE protein to induce the double knockout (DKO) in vitro (Fig. 5A and Supplementary Fig. 27). We performed tcBS-seq with the DKO and control lines, which allowed us to determine the methylation level for ~2M CpGs (9% of CpGs in the mouse genome). As shown previously[34], we found that methylation levels are globally unchanged between the control and *Dnmt3a/3b* deficient MEFs (Fig. 5B). To determine whether methylation levels vary at individual CpG sites, we calculated the difference in methylation between the DKO and controls (Fig. 5C and D). There was no consistent depletion amongst the different methylation states (U, I, and M). Indeed, between the DKO and control cell lines, fewer than 3000 CpGs (~0.14%) show significantly different levels of methylation ($q < 0.01$; Fisher's exact test), with about half increasing and the other half decreasing in methylation level. Additionally, an ANOVA analysis revealed that the independent control cell lines (A–D) contribute substantially more to the variance in methylation ($F = 60.7$, $p < 2e{-16}$) than the knockout condition ($F = 1.6$, $p = 0.20$). These results show that the deposition of neither probabilistic nor faithful methylation is dependent on DNMT3A/3B in somatic cells, and instead suggest that DNMT1 is the sole methyltransferase involved in both processes.

## Discussion

Based on the premise of its faithful clonal inheritance between cell divisions and its potential to influence transcription, DNA methylation is frequently used as a biomarker for epigenetic ageing and in clinical and epidemiological studies[35–39]. Our results show that intermediately methylated CpGs are unfaithfully and probabilistically propagated during clonal expansion with implications for the use of methylation as a biomarker. We find that these intermediately methylated loci are generally associated with a lack of gene expression, meaning that any functional interpretations are also likely to be unreliable. Due to the observed relationship between gene expression and methylation fidelity, it is important to consider that intermediate methylation

states may vary between cell types in accordance with transcriptome-wide fluctuations.

For the purposes of evaluating methylation fidelity, we established a framework that only considers symmetric methylation states because hemimethylation is simply unfaithful according to the semi-conservative model of methylation inheritance. However, we expect that some, if not many, of the intermediately methylated CpGs reflect a heterogenous mixture of both symmetrically methylated and hemi-methylated sites. Both mass spectrometry and sequencing-based approaches revealed that post-replicative methylation deposition is not immediate, yielding a hemimethylated landscape, with early-replicating regions of the genome regaining methylation more quickly compared with late-replicating ones[40–42]. Because our experimental design primarily probes early-replicating regions of the genome, it is unlikely that the unfaithful methylation inheritance we observed is due to differences in post-replicative deposition dynamics.

The probabilistic, rather than stochastic, nature of methylation changes at intermediate sites implies that, despite the unfaithfulness, there is a weighted directionality of methylation inheritance during clonal expansion that is reliant on an element of chance. For example, if a CpG is 70% methylated in the parental population of cells, it is more likely to gain and retain methylation during clonal expansion, while a CpG that is 30% methylated has a higher likelihood of losing methylation. We suggest that the probabilistic gain in methylation is likely due to the de novo function of DNMT1[43–46], while the mechanism for the loss is still unclear. Additionally, differential occupancy of transcription factors could influence the dynamics of methylation deposition at intermediately methylated regions, giving rise to probabilistic methylation states[7]. Although we could not assign functional features to the intermediately methylated sites we identify, such functionality cannot be ruled out. Our findings challenge the long-standing assumption in the epigenetics field that DNA methylation is a mitotically inherited modification in somatic lineages by revealing that at intermediately methylated sites, methylation levels are probabilistically, not clonally, maintained within a cell population.

## Methods

### Mouse lines

Mouse work was conducted under project licences from the UK government Home Office (project license numbers: PC9886123, PC213320E, and PP8193772). Mice were housed in a temperature and humidity-controlled room under 12 h light/12 h dark cycles and fed a standard chow diet ad libitum. Post-implantation embryos (E13.5 for mouse embryonic fibroblasts) and blastocysts (E3.5 for mouse embryonic stem cells) were collected by natural mating, and the plugged date of conception was considered E0.5. *Dnmt3a^flox/flox^3b^flox/flox* mice[47] were obtained from RIKEN BioResource Research Center (BRC) and maintained on a C57BL/6 mouse background.

### Cell line generation and culture

Mouse embryonic fibroblasts (MEFs) were established from E13.5 C57BL/6J mouse embryos[48]. MEFs were grown at 37 °C and 5% CO₂ in high glucose DMEM GlutaMAX™ (ThermoFisher, cat no. 31966021) supplemented with 10% FBS (Gibco, 11550356) and 1% penicillin/streptomycin solution and passaged with trypsin/EDTA. MEFs were immortalised by serial passaging the cells through the crisis phase[49]. Isolation of single immortalised MEF cells was performed by flow cytometry (MoFlo Astrios Cell Sorter) to 96-well plates. The isolated single cells were grown into clonal cell lines by successive passaging from 96-well, 48-well, 24-well, 12-well, and 6-well plates to 10 cm dishes where they were grown to 75% confluency to yield 1–2 million cells per clonal line for DNA/RNA extraction. The parental lines, from which the clonal lines were derived, were also grown to 75% confluency on 10 cm dishes to yield 1–2 million cells for DNA/RNA extraction.

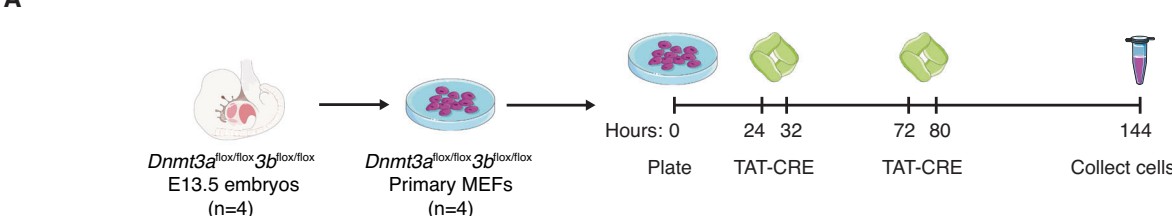

**A**

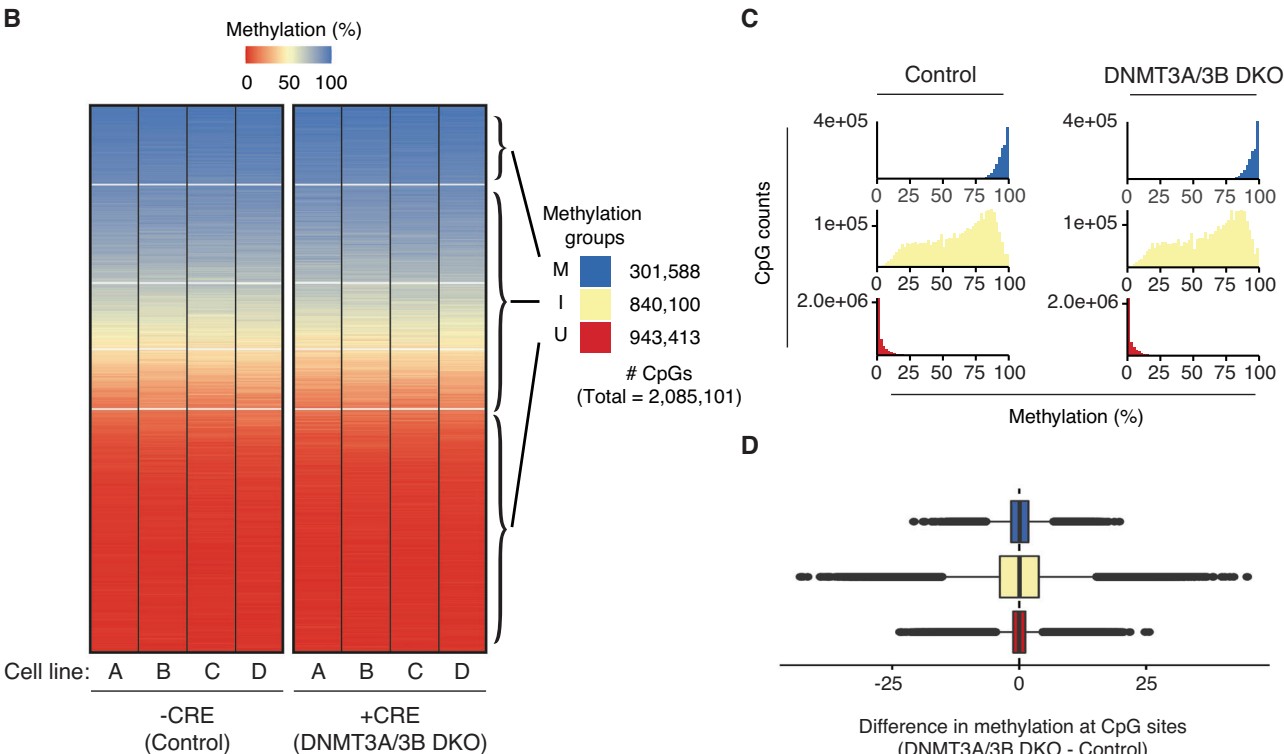

**Fig. 5 | Intermediate methylation does not arise due to *DNMT3a/3b* de novo activity. A** Primary MEFs were established from *Dnmt3a*$^{flox/flox}$*3b*$^{flox/flox}$ E13.5 embryos (*n* = 4). The cells were then plated and 24 h later, treated with recombinant TAT-CRE protein for 8 h. 48 h after the initial treatment, the cells were treated with TAT-CRE again for 8 h. 72 h after the second TAT-CRE treatment, the cells were collected for DNA/RNA extraction. We generated tcBS-seq libraries for four individual control MEF lines and four matched TAT-CRE treated MEF lines. **B** Heatmaps of 2,085,101 CpGs sorted by median methylation (%) within *k*-means clusters (*k* = 5) for untreated (control) and TAT-CRE treated (DNMT3A/3B) primary MEF cell lines A–D. Methylation groups are classified using the *k*-means clusters as shown. U (red) = consistently unmethylated across all the cell lines. I (off-white) = potential to be intermediately methylated. M (blue) = consistently methylated across all the cell lines. **C** Methylation percentage distributions of the methylation groups for both the control and DNMT3A/3B DKO cell lines. **D** Boxplot showing the difference in methylation at CpG sites between DNMT3A/3B DKO and control cell lines. Differences per CpG were calculated as the difference in average methylation across the four DNMT3A/3B DKO cell lines and the four control cell lines at a given CpG site. U (red; *n* = 943,413); I (off-white; *n* = 840,100); M (blue; *n* = 301,588). For **D**, the box of the boxplot shows the 25th, 50th, and 75th percentiles; the whiskers extend to 1.5*IQR beyond the edges of the box (where IQR = 75th–25th percentile), with outliers shown as dots.

## DNA/RNA extraction

DNA/RNA extraction was performed with the Qiagen AllPrep DNA/RNA Mini Kit (cat no. 80204).

## Target capture bisulfite sequencing (tcBS-seq)

Libraries were generated using the SureSelectXT Methyl-seq Library Preparation Kit with the following specifications. 1 µg of DNA was sonicated to an average size of 200 bp with the Covaris E220 (duty factor = 30%, PIP = 100, cycles per burst = 1000, treatment time = 95, bath temperature = 7 °C, with intensifier fitted) using 50 µl microTUBEs (cat no. 520166) and 24-place rack (cat no. 500308). Following end repair, dA tailing, and adaptor ligation, libraries were hybridised to single-stranded RNA probes homologous to 297,000 regions in the mouse genome (SureSelectXT Methyl-Seq Capture Probes, cat no.

931052). After purification, the libraries were bisulfite converted using Zymo Research's Methylation-Gold Kit (cat no. D5005) and PCR amplified for 8 cycles. The PCR-amplified bisulfite-treated libraries were then purified, indexed by PCR amplification (6 cycles), and purified again. All purification steps took place using Ampure XP beads (Beckman Coulter, cat no. A63881). The final libraries were then quality-checked and quantified for multiplexing using the Bioanalyzer High-Sensitivity DNA kit (Agilent, 5067-4626) and Qubit dsDNA HS Assay kit (Thermo Scientific, Q32854). The multiplexed libraries were sequenced as 150 bp paired-end reads on the Illumina Novaseq 6000 SP. The resulting tcBS-sequencing data was trimmed by Trim Galore (v0.6.0) and aligned to mm10 using Bismark[50]. Reads with map quality score <10 were excluded and were further filtered by M-bias filtering[51]: for MEF-1 and MEF-2 data, we excluded the first two bp on both paired

reads and the last bp on Read 2; for MEF DNMT3A/3B DKO and control data, we excluded the first two bp on Read 1 and the first four and last two bp on Read 2. Methylation data was extracted using bismark_methylation_extractor[50] and analysed in R (v3.6.1).

## Filtering and thresholding tcBS-seq methylation data

The SureSelectXT tcBS-seq system uses RNA probes homologous to 300,000 genomic regions, which represent 109 Mbases of the 2.7 Gbase mouse genome, and about three million CpGs of the total 20 million CpGs. We use read coverage across the individual sequencing libraries to look for enrichment of reads on individual chromosomes to determine coverage-based karyotypes for our immortalised MEF cell lines and find that chromosomes 12, 18, and 19 exhibits aneuploidy in at least one of the cell lines (Supplementary Fig. 1A and B). Mean or median coverage for both the MEF-1 and MEF-2 cell lines was determined for a single CpG, then normalised by the sum of mean or median coverages and plotted as the negative log2 to more easily visual the read coverage across chromosomes—i.e. lower values, therefore, represent higher coverage, and vice-versa.

For subsequent analyses, we remove the CpGs on the aneuploid chromosomes, as well as those on the X chromosome due to the sex difference between the two parental MEF lines (Supplementary Fig. 1C). We used the following primers to amplify the SRY gene on the Y chromosome:

SRY_F: GCAGGCTGTAAAATGCCACT
SRY_R: TTCCAGGAGGCACAGAGATT

After thresholding for CpGs with ≥10x coverage in all 16 sequencing libraries, for subsequent analyses, we retain 1.2 million CpGs (or ~5% of CpGs in the mouse genome) with a median coverage of 32 reads per CpG per dataset (Supplementary Fig. 1D).

To assess whether the tcBS-seq CpGs are representative of CpGs genome-wide, we examine the distribution of CpGs amongst genomic annotations and compare the methylation data to similar whole-genome bisulfite methylation datasets. First, we calculate the proportion of various genomic annotations covered by at least one tcBS-seq CpG. We find that these CpGs are found within 27.3% of promoters, 13.8% of exons, and 22.9% of introns in the genome, whereas they only overlap with ~1–2% of transposable elements (TEs) (Supplementary Table 2). This informs us that tcBS-seq enriches for genic regions and is depleted for TEs.

Next, we compare the methylation profiles of all MEF-1 ($n = 8$) and MEF-2 ($n = 8$) tcBS-seq datasets to other relevant methylation data (Supplementary Table 3). To do this, we utilised publicly available WGBS datasets from both in vitro (primary and immortalised MEFs)[52,53] and in vivo contexts (E13.5 and E14.5 embryonic limb bud3)[54]. When filtered for the same CpGs covered in our dataset, we observe that all the datasets have similar methylation distributions with enrichment for hypo- and hypermethylation, and depletion of intermediate states (Supplementary Fig. 2A). We observe that the MEF-1, MEF-2, and publicly available primary MEF datasets, are globally reduced in methylation compared to immortalised MEFs and E13.5/E14.5 embryonic limb bud (Supplementary Fig. 2B). However, when considering the entire genome, the tcBS-seq MEF-1, and MEF-2 datasets are depleted in global methylation levels compared to the WGBS datasets. This suggests that MEF-1 and MEF-2 methylation profiles are more like those of primary MEFs as opposed to immortalised MEFs, and more importantly, that tcBS-seq enriches for hypomethylated regions of the genome.

Different kinds of regions in the genome exhibit distinctive methylation patterns. For example, genomic imprints are allelically methylated and exhibit 50% methylation levels, while TEs are hypermethylated compared to the background methylation level of the genome. Additionally, gene promoters are generally hypomethylated, whereas exons and introns tend to be hypermethylated. We compare methylation profiles of the relevant public datasets at imprints, gene

bodies, and SINEs (a major family of TEs), to validate that, despite being depleted for methylated regions, the MEF-1 and MEF-2 datasets show the expected distinctive methylation distributions. We filter publicly available methylation datasets for the tcBS-seq-covered CpGs and find that all the datasets have similar distributions of methylation at imprints, across gene bodies, and at SINEs (Supplementary Fig. 2C–E). This suggests that despite the underrepresentation of TEs and methylated regions in the tcBS-seq, there is enough methylation data that is representative and typical of the somatic mouse methylome to address questions regarding methylation inheritance through clonal expansion.

## Characterising the clonal methylation data

Here we explain the calculations for the data visualised by the heatmaps of Fig. 1D–F. CpG density was calculated as the number of CpGs within 100 bp of the focal CpG, with an upper limit of 30 CpGs. Fidelity score was calculated as the number of clonal lines that exhibit [0–10]%, [40–60]%, or [90–100]% methylation, divided by 14 (the total number of clonal lines). Neighbour similarity score was calculated as the number of clonal lines in which the closest CpG to a focal CpG is within 10% methylation, divided by 14 (the total number of clonal lines).

## Classifying CpGs by methylation

Throughout this manuscript we classify CpGs in five different ways for subsequent analyses: (1) As $k$-mean clusters, (2) combined $k$-means methylation groups, (3) methylation state bins, (4) probabilistic and faithful, and (5) combined $k$-means methylation groups for the conditional DNMT3A/3B knockout experiment.

1. Figure 1C shows the 7 $k$-means clusters of methylation data arranged by median methylation.
2. Supplementary Fig. 3A and B show how we combine the $k$-means clusters to unmethylated (U), unmethylated and intermediately methylated (UI), intermediately methylated (I), methylated and intermediately (MI), and methylated (M).
3. For Fig. 4B, C and Supplementary Fig. 25A and B, we characterise CpG methylation states as low [0, 10), low intermediate [10, 40), intermediate [40, 60), high intermediate [60, 90), and high (90, 100].
4. For Supplementary Fig. 26, we define CpGs as probabilistic if a clonal line exhibits (10–40]% or (60–90]% methylation amongst both the MEF-1 and MEF-2 clonal lines. CpGs are defined as faithful if all clonal lines exhibit [0–10]%, (40–60]%, or (90–100]% methylation.
5. Fig. 5B and C show how we combine the $k$-means clusters to unmethylated (U), intermediately methylated (I), and methylated (M).

## Identifying unfaithful and faithful regions of the genome

Unfaithful regions (≥5 bp) were identified with an average fidelity score < 0.75 across at least 3 CpGs. Faithful regions (≥5 bp) were identified with an average fidelity score = 1 across at least 3 CpGs. methylation. Methylation calls from the original top and original bottom strands were combined from the Bismark methylation extractor v0.20.0 output files. Only reads with MAPQ of at least 10 were considered. CpG sites having fewer than 10 methylation calls across the samples in each cell line were excluded, and reads having fewer than 3 covered CpGs were excluded. Reads were then ordered by average methylation of covered CpG sites.

## Analysing methylation at transposable elements

To estimate the proportion of TEs for which methylation data is available (Supplementary Table 2), CpG coordinates were compared with the RepeatMasker v4.1.1 annotation for mm10 (obtained from the UCSC genome browser). Data on transposon sequence divergence from consensus was taken directly from RepeatMasker output and

plotted for each transposon class (i.e., DNA transposons, LTR retrotransposons, LINEs, and SINEs). Consensus alignment positions (1–200 from 5′-ends) for genomic copies of SINEs were similarly extracted from the mm10 RepeatMasker output.

## Total RNA sequencing

Libraries were generated using the NEBNext® rRNA Depletion Kit (NEB, cat no. E6310) and Ultra™ II Directional RNA Library Prep Kit for Illumina® (NEB, cat no. E7760) with the following specifications. RNA Integrity Number (RIN) was determined using the Agilent RNA 6000 Pico Kit (cat no. 5067-1513) on the Agilent 2100 Bioanalyzer and its associated software. 1 µg of RNA was hybridised to probes for rRNA depletion, treated with RNase H and DNase I, and purified before being fragmented at 94 °C with incubation times ranging from 8 to 15 min depending on the RIN. The fragmented RNA was then reverse-transcribed to cDNA in two steps and purified. Following end repair and adaptor ligation, the cDNA libraries were purified, PCR enriched for 9 cycles and purified again. All purification steps took place using Ampure XP beads (Beckman Coulter, cat no. A63881). The final libraries were quality checked and quantified for multiplexing using the Bioanalyzer High-Sensitivity DNA kit (Agilent, 5067-4626), Qubit dsDNA HS Assay kit (Thermo Scientific, Q32854), and the KAPA Library Quantification Kit optimised for Roche® LightCycler 480 (Roche, 07960298001). The multiplexed libraries were sequenced as 150 bp paired-end reads on the Illumina Novaseq 6000 S1. The resulting RNA-sequencing data was trimmed by Trim Galore (v0.6.0), aligned (mm10) and quantified using Salmon (v1.5.2)[55] using the mm10 reference annotation (Ensembl release 102, November 2020). The transcript quantification was processed using the R/Bioconductor package DESeq2 (v1.24.0)[56], to obtain normalised counts using the "regularised log" (rlog) transformation.

## Characterising transcriptomic data

For transcriptomic analyses, only genes annotated as protein-coding by Ensembl (release 102) were considered, and single-exon genes were excluded, for a total of 20,273 genes. To get a single transcript per gene, canonical transcripts were first defined as the most highly expressed transcript from a gene on average across all the MEF parental and clonal total RNA-seq datasets; for genes lacking transcript expression in those datasets, the mm10 "known canonical" transcripts as defined by the UCSC genome browser were used. From these single transcripts, the "first exon" for each gene was determined; the promoter region was defined as 1000 bp prior to this first exon. The annotations were classified into the following genic regions: promoters, 1st exons, 1st introns, 2nd exons, 2nd introns, 3rd exons, 3rd introns, rest of exons, rest of introns, last introns, last exons (Supplementary Table 4). Only genic regions covered by at least three CpGs in the methylation data and >6 bp in length were considered. Next, each genic region was divided into five tiles. Transcription quintiles were derived from the normalised expression values for each annotated protein-coding gene averaged across all the MEF clonal and parental total RNA-seq datasets and assigned to the corresponding tiled regions of the genes (Fig. 3A, B and Supplementary Figs. 20–22). CpGs were overlapped with protein-coding transcripts and their promoters (1000 bp prior to the TSS of each of the transcripts) and assigned a corresponding transcription quintile (Fig. 1D and Supplementary Fig. 21B). CpGs that did not overlap with a protein-coding transcript or a promoter, were classified as intergenic (Fig. 1D and Supplementary Fig. 21C).

## In vitro knockout of *Dnmt3a/3b*

Primary MEFs were established from *Dnmt3a*^flox/flox^*3b*^flox/flox^ E13.5 embryos[48] and grown at 37 °C and 5% $CO_2$ in high glucose DMEM GlutaMAX™ (ThermoFisher, cat no. 31966021) supplemented with 10% FBS (Gibco, 11550356) and 1% penicillin/streptomycin solution and

passaged with trypsin/EDTA. To induce the knockout of DNMT3A/3B, $1 \times 10^6$ cells were plated and the next day washed with PBS and the media was replaced with 5 ml of DMEM:PBS (1:1) with 50 µl TAT-CRE recombinase (2–3 µg/µl) and incubated at 37 °C for 8 h. Following the TAT-CRE recombinase treatment, cells were washed three times with PBS and the original growth media was replaced. The TAT-CRE recombinase treatment was carried out again 48 h following the start of the initial treatment. Cells were collected for protein and DNA/RNA extraction 72 h after the second TAT-CRE recombinase treatment. Knockout of the genes was confirmed by running PCR reactions using Q5® High-Fidelity DNA Polymerase (NEB, cat no. M0491) and the following primers on a 2.5% agarose gel:

Dnmt3a_Ex17_F: AGATCATGTACGTCGGGGAC
Dnmt3a_intron19_R: AGACAAGACAGGGACGAAGC
Dnmt3b_Ex16_F: ATGCTTCTGTGTGGAGTGTCTGG
Dnmt3b_intron20_R: AGGGGTCACAAAACACAGGT

## Western blotting

Flash frozen control and knockout MEFs were thawed on ice and lysed with RIPA buffer (Sigma-Aldrich, cat no. R0278), supplemented with EDTA-free cOmplete™ Protease Inhibitor Cocktail (Roche), for 20 min on ice and centrifuged 14,000×g for 10 min at 4 °C. Supernatant was collected and protein concentrations were determined using Bradford Reagent (Sigma-Aldrich, cat no. B6916). Protein samples were boiled with 4x Laemmli sample buffer (Bio-Rad, cat no. 1610747), with 10% b-mercaptoethanol, at 95 °C for 5 min. Protein ladder (Thermo Fisher Scientific, cat no. 26619) and equal amounts of protein sample were then loaded onto a 4–20% gradient SDS–PAGE gel (Bio-Rad, cat no. 4561095) and run at 120 V before being transferred to a PVDF membrane (Bio-Rad, cat no. 1704156). The membrane was then blocked with 5% skim milk and incubated with the following primary antibodies for 1 h at room temperature: anti-DNMT3a (1:1000, ab188470) and anti-b-actin (1:1000, ab8227). After three 10 min washes with 1xTBST buffer at room temperature, the membrane was incubated with a secondary antibody, goat anti-rabbit IgG-HRP (1:3000, Agilent, cat no. P044801-2), for 1 h at room temperature. The membrane was again washed three times for 10 min at room temperature with 1xTBST buffer; the signal was detected using Amersham ECL (GE Healthcare, cat no. RPN2232) and imaged on an LI-COR Odyssey® Fc Imaging System.

## Reporting summary

Further information on research design is available in the Nature Portfolio Reporting Summary linked to this article.

## Data availability

All raw and processed sequencing data generated in this study have been submitted to the NCBI Gene Expression Omnibus (GEO; https://www.ncbi.nlm.nih.gov/geo/) under accession number GSE234695 (parental/clonal tcBS-seq and total RNA-seq; *Dnmt3a/3b* DKO tcBS-seq) and to the Sequence Read Archive (SRA, https://www.ncbi.nlm.nih.gov/sra) under accession number PRJNA980423 (*Dnmt3a/3b* DKO RNA-seq).

## Code availability

All code used to perform the analyses is available on GitHub (https://github.com/AFS-lab/methylation_fidelity; https://doi.org/10.5281/zenodo.8151496).

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

## Acknowledgements

We thank members of the Ferguson-Smith group for technical support and useful discussion. We thank Tessa Bertozzi and Ben Simons for critical reading of the manuscript. We thank Mitsuteru Ito for technical assistance. We thank Rahia Mashoodh for support with statistical analysis. We are grateful to Yoach Rais for providing us with TAT-CRE recombinase. We thank Ben Harvey and Joe Gaughan from Agilent for technical support and valuable input. This work was supported by grants from the Wellcome Trust Investigator Award 210757/Z/18/Z (to A.C.F.-S.), MRC Programme Grant MR/R009791/1 (to A.C.F.-S.), Wellcome Trust and Royal Society Sir Henry Dale Fellowship 206257/Z/17/Z (to F.K.T.), Human Frontier Science Programme CDA-00032/2018 (to F.K.T.) Cambridge Trust International Studentship (to A.D.H.), and Walter Benjamin Postdoctoral Fellowship from the Deutsche Forschungsgemeinschaft (to D.G.). Parts of Figs. 1A and 5A were drawn by using pictures from Servier Medical Art. Servier Medical Art by Servier is licensed under a Creative Commons Attribution 3.0 Unported License (https://creativecommons.org/licenses/by/3.0/). For the purpose of Open Access, the author has applied a CC BY public copyright license to any Author Accepted Manuscript (AAM) version arising from this submission.

## Author contributions

A.D.H., F.K.T., and A.C.F.-S conceived and designed the project. A.D.H. performed the experimental work and bioinformatics analyses. N.J.K. contributed to the methylation data processing and analyses. D.G. contributed to the analysis of transposable elements. H.T. contributed to the RNA-sequencing processing and analysis. N.T. contributed to the inducible knockout mouse work. A.D.H., F.K.T., and A.C.F.-S drafted the manuscript. All authors commented on the manuscript.

## Competing interests

The authors declare no competing interests.
