## [Peer Review File · Nature Communications]

Epigenetic inheritance is unfaithful at intermediately methylated CpG sitesREVIEWER COMMENTS

Reviewer #1 (Remarks to the Author):

The submitted manuscript can best be described as “provocative” – in the best sense of the word. Canonical concepts in epigenomics paint a simple picture in which a methylated CpG is maintained in that state through mitosis via the action of DNMT1. In those cases where the methylation is erased (eg in resetting imprints), this takes place in a deterministic, targeted way. The manuscript presents a compelling dataset that – in a specific set of settings – this is not the case. Using a conceptually simple and cogent experimental setup – two parental fibroblast lines derived from the same embryo, and 6 clonal lines each derived from these – the manuscript profiles CpG “fidelity” across ca 5% of the mouse methylome and finds that regions of intermediate methylation (which happen to coincide with poorly transcribed regions) are strikingly “unfaithful” in their maintenance. Further, it appears to be not a “haplotype-based” phenomenon (ie a patch of CpGs, such as a CpG island, exhibiting concordant “unfaithful” behavior), but rather a CpG-by-CpG one. This is surprising and noteworthy.

What is the biological impact of this (in simple terms – why do we care?)? The authors state it well:

“This suggests that either transcriptional activity defines the patterns of faithful hypo- and hypermethylation throughout a gene, or that a faithful methylation state contributes to expression. Furthermore, it indicates that the absence of transcription may be permissive for the presence of unfaithful intermediate methylation levels.”

The former hypothesis and the last sentence can be restated as: if a locus is not transcribed, Mother Nature stops “caring” about CpG methylation which then starts to drift in stochastic ways. This is good to know, but not particularly exciting. The latter hypothesis is, in contrast, quite exciting, because it suggests that active methylation homeostasis programs may be targeted in cell-type-transcriptome-specific-ways. The authors’ finding that the quasi-stochastic gain-and-loss of CpG methylation they observe is not reliant on the de novo MTases (DNMT3a/b), but likely driven by DNMT1 alone, is interesting and points to one avenue of study that could distinguish between the two models. Specifically, given the epigenomic histone marks of repressive chromatin that “track” with unfaithful methylation, the possibility exists that such marks create domains that are somehow restrictive for DNMT1 action (in maintaining existing DNA methylation, or inscribing the mark de novo). All the above are “comments on the margins” of the submitted manuscript meant to point out noteworthy ramifications of their findings (which thus make the work interesting even further).

Comments

Fig 1c presents a compelling metaanalysis of methylation level, and fidelity of its inheritance. It would potentially be even more compelling if the authors could present (using the traditional scheme of “lollipops” with either white or black circles) a few genomic loci that illustrate the different behaviors with respect to methylation fidelity. In such a figure, a representative locus that exhibits high methylation fidelity between the two MEF lines, and across their 6 clonal progeny, would one presume show invariant methylation patterns. In contrast, loci that are unfaithful in maintain the methylated (or demethylated) state would show differences between the two MEF lines or (which is even more interesting), between the 6 clonal lines.

Do the authors think that PEV could be a relevant phenomenon here? Specifically, is there an opportunity to use their data to identify loci that are at the interface between faithful (active gene, demethylated), and unfaithful (silent gene, variegating methylation) states? Could there be, eg, known boundary elements between the two (mouse ENCODE has a wealth of data on this)? The authors are well-familiar with the many systems on PEV (from the classic Muller one to various subtelomeric systems, and mammalian PEV, eg Emma Whitelaw’s work among others), the classical studies on the agouti locus etc.

It is unclear from the data whether there is a directionality to the unfaithfulness? In regions that

are “unfaithful” – is it more likely that a methylated CpG becomes unmethylated, or the other way around? Or are both directions equally likely?

//Fyodor Urnov, Innovative Genomics Institute, UC Berkeley//

Reviewer #2 (Remarks to the Author):

NCOMMS-23-10317-T Review

SUMMARY

DNA methylation, an important epigenetic mark, is found on the majority of cytosines in CpG dinucleotides in mammalian genomes, where it most often symmetrically marks the two cytosines on complementary DNA strands. Following DNA replication, the resulting hemi-methylated state is recognized by and recruits the methyltransferase machinery to restore symmetric methylation and maintain the methylome between cell divisions. When assaying methylation by next-generation sequencing, using millions of cells as input, the majority of CpGs have either fully unmethylated or fully methylated readouts, indicating high homogeneity in methylation state at these loci. However, a subset of loci reliably give intermediate methylation readings, which could represent multiple possible scenarios. This is a broadly observed phenomenon, and understanding the source of intermediate methylation is of high significance, as doing so could inform on both the distribution of DNA methylation in populations and the fidelity of its propagation during mitosis.

One prevalent theory is that intermediately methylated loci represent intrapopulation heterogeneity- some cells are fully methylated and propagate this state to any and all progeny, while others are unmethylated and similarly propagate the unmethylated state. In this scenario, the methylation machinery faithfully restores all methylation to the fully symmetric state post-replication, and the intermediate methylation observed is a consequence of one population of cells being methylated and the other unmethylated.

To test this, Hay and colleagues created 2 immortalized MEF cell lines, from each of which 7 single cells were sorted and expanded to create clonal cell lines. The authors then did large-scale, targeted bisulfite sequencing (tcBS-seq) to compare methylation between parental and clonal lines. They did RNA-seq to compare methylation and expression, and additionally profiled Dnmt3a-3b DKO MEFs using tcBS-seq to assess the contribution of the de novo methyltransferases to intermediate methylation. The experimental design is elegant and straightforward, and the analyses presented are well-done and illustrated very nicely in the figures.

Based on their tcBS-seq data in clonally expanded MEFs, the authors conclude that intermediately methylated CpGs represent loci where DNA methylation is unfaithfully inherited between cell divisions. However, there are alternative explanations for the results that are not discussed. The authors repeatedly claim that their data tracks methylation inheritance between cell divisions, when their experiments do not track inheritance through mitosis or make any comparisons based on cell division/ cell cycle stage. The text should make this explicit and frame the experiments described as measuring population heterogeneity following clonal selection, which is itself an interesting and informative experimental design. To exclude the possibility that intermediate methylation is a consequence of delayed DNA methylation maintenance following replication, the frequency and distribution of intermediate CpGs in cycling and arrested cells should also be compared (see Major Points).

Major Points

Requested Experiments and Analyses

1. The authors conclude that their observations are of methylation inheritance between cell divisions, but do no analysis of cell cycle dynamics or replication timing. Given that methylation is

known to be less stable/ more readily lost in late-replicating DNA, it would be informative to compare the distribution of the methylation categories defined in Figure 1C with replication timing. Replication timing domains for MEFs have already been defined (e.g., DOI: 10.1101/gr.099796.109, DOI: 10.1007/s10577-022-09703-7).

2. In a similar vein, the Meissner lab has shown that blocking a cell population in mitosis reduces intermediate methylation (DOI: 10.1038/s41594-018-0046-4), suggesting that intermediately methylated CpGs in both parental and clonal MEF lines could represent loci that are slower to re-methylate post-DNA replication, rather than loci that differ in methylation between cells in the same cell cycle stage. To elucidate the role of DNA replication/ the cell cycle in the detection of intermediately methylated CpGs, it would be important to compare cycling and arrested populations by the tcBS-seq method used here. It would also be informative to show the cell cycle distribution of the MEF-1 and MEF-2 cell lines.

3. Did anything distinguish the CpGs that stably differed in methylation status between MEF-1 and MEF-2? And, did anything distinguish the CpGs that differed between parental and clonal lines? How many such CpGs were identified?

4. The authors use an enrichment-based method, tcBS-seq, rather than an unbiased whole-genome approach. Can some characterization of the probes used (how representative they are of the genome regarding CpG density, genic/intergenic/TE proportions etc) be included, to better contextualize the loci studied here?

Requested Edits in Text

5. As depicted in Figure 1, the authors only consider symmetric methylation states in their interpretation of the data. However, the tcBS-seq method wouldn't be able to discriminate monoallelic methylation from biallelic hemi-methylation, and a single cell could also conceivably have 25% or 75% methylation at a single locus if one allele is asymmetrically methylated and the other is symmetrically methylated/ unmethylated. Whether or not hemi-methylated CpGs in any of these contexts (originating from DNA replication, or de novo methylation gain/loss on a single DNA strand) could underlie intermediate methylation at a population level should be discussed.

6. In this work, population heterogeneity following clonal selection is analyzed to gain insights into the faithfulness of methylation propagation in cycling cells. However, several groups have now directly analyzed methylation on replicated DNA over time by sequencing or mass spectrometry. These experiments assess mitotic inheritance of DNA methylation, and should be included in the cited literature to contextualize the findings of this study.

7. Can the authors comment on the selection of MEFs for this study, given that more arguably homogeneous cell types (e.g., mESCs) also have well-documented regions of intermediate CpG methylation?

8. The Methods and Extended Data Tables 2 and 3 refer to cultured mESC/ E3.5 data that isn't presented in any of the figures, so these references should be removed.

9. Regarding Extended Data 10, it is important to point out that though H3K27me3 and H3K9me3 are both associated with repression, H3K27me3 tends to anticorrelate with DNA methylation and H3K9me3 positively correlates with it. The H3K9me3 association with intermediate methylation is interesting, given that direct quantification of DNA methylation on DNA pulled down with H3K9me3-marked nucleosomes shows very high methylation levels (DOI: 10.1038/s41556-022-01048-x). Could these be intermediate CpGs as a consequence of their residing in late-replicating regions?

Minor Points

10. Is the text describing relative genome coverage correct in the Extended Data Fig 1 legend? Based on the figure MEF-1 (red) has increased coverage at chr12/18/x and decreased coverage at

chr19, but this is flipped in the text.

11. In Figures 1G and 1H, it seems that individual CpGs can both have an expression designation and be intergenic. Since the expression is over protein-coding genes, shouldn't these be mutually exclusive?

12. Extended Data Tables 1 and 2 don't seem to relate to the text in which they're referenced in the Results section.

13. In Extended Data 2E, could the difference in 3T3 MEFs compared to the other datasets be briefly commented on?

14. In the Main Text, paragraph 3, please cite the panels being discussed in the second half of the paragraph for clarity.

15. In Extended Data 3C-E, could the datasets being compared statistically please be depicted more clearly in the figures?

16. In Extended Data 4A, there's a mix of brackets and parentheses that should be standardized.

17. In Figure 2A/B and Extended Data 6A/B, is the interquartile range only shown for the "none" and "high" expression categories? This is how it appears in the graphs, but should be made clear if so in the legends.

18. The text referencing Extended Data 8 indicates that the figure depicts the proportion of TEs in exons, but it does not.

19. The key in Figure 3B should be copied into Extended Data Figure 9A to help with interpretation.

20. In the legends for Figure 3C and Extended Data 9B, it should be stated what the green dots and lines in the UpSet plot represent, just for completeness.

21. The discussion of statistics and trends in the DKO data could be more clear as a figure, rather than being solely in the text.

Thank you to the reviewers for their comments and suggestions as they have allowed us to sharpen, improve, and clarify the manuscript. We have responded to each of the reviewer's comments below with green coloured text and have referred to any subsequent changes in the text or figures.

REVIEWER COMMENTS

Reviewer #1 (Remarks to the Author):

The submitted manuscript can best be described as “provocative” – in the best sense of the word. Canonical concepts in epigenomics paint a simple picture in which a methylated CpG is maintained in that state through mitosis via the action of DNMT1. In those cases where the methylation is erased (eg in resetting imprints), this takes place in a deterministic, targeted way. The manuscript presents a compelling dataset that – in a specific set of settings – this is not the case. Using a conceptually simple and cogent experimental setup – two parental fibroblast lines derived from the same embryo, and 6 clonal lines each derived from these – the manuscript profiles CpG “fidelity” across ca 5% of the mouse methylome and finds that regions of intermediate methylation (which happen to coincide with poorly transcribed regions) are strikingly “unfaithful” in their maintenance. Further, it appears to be not a “haplotype-based” phenomenon (ie a patch of CpGs, such as a CpG island, exhibiting concordant “unfaithful” behavior), but rather a CpG-by-CpG one. This is surprising and noteworthy.

What is the biological impact of this (in simple terms – why do we care?)? The authors state it well:

“This suggests that either transcriptional activity defines the patterns of faithful hypo- and hypermethylation throughout a gene, or that a faithful methylation state contributes to expression. Furthermore, it indicates that the absence of transcription may be permissive for the presence of unfaithful intermediate methylation levels.”

The former hypothesis and the last sentence can be restated as: if a locus is not transcribed, Mother Nature stops “caring” about CpG methylation which then starts to drift in stochastic ways. This is good to know, but not particularly exciting. The latter hypothesis is, in contrast, quite exciting, because it suggests that active methylation homeostasis programs may be targeted in cell-type-transcriptome-specific-ways. The authors’ finding that the quasi-stochastic gain-and-loss of CpG methylation they observe is not reliant on the de novo MTases (DNMT3a/b), but likely driven by DNMT1 alone, is interesting and points to one avenue of study that could distinguish between the two models. Specifically, given the epigenomic histone marks of repressive chromatin that “track” with unfaithful methylation, the possibility exists that such marks create domains that are somehow restrictive for DNMT1 action (in maintaining existing DNA methylation, or inscribing the mark de novo). All the above are “comments on the margins” of the submitted manuscript meant to point out noteworthy ramifications of their findings (which thus make the work interesting even further).

Comments

Fig 1c presents a compelling metaanalysis of methylation level, and fidelity of its inheritance. It would potentially be even more compelling if the authors could present (using the

traditional scheme of “lollipops” with either white or black circles) a few genomic loci that illustrate the different behaviors with respect to methylation fidelity. In such a figure, a representative locus that exhibits high methylation fidelity between the two MEF lines, and across their 6 clonal progeny, would one presume show invariant methylation patterns. In contrast, loci that are unfaithful in maintain the methylated (or demethylated) state would show differences between the two MEF lines or (which is even more interesting), between the 6 clonal lines.

This is an excellent suggestion and we thank the reviewer. In response, we have generated illustrative “lollipop” plots for unfaithful, faithful (hyper- and hypomethylated), and imprinting regions for MEF-1 and MEF-2. As predicted by the reviewer, these plots provide a very compelling way to illustrate the data and are likely to resonate with most people working in the DNA methylation field. Therefore, we have included four of these plots showing each region type for MEF-1 as a new main Fig. 2 (p. 4-5, lines 164-169; p. 14) – subsequently renumbering Fig. 2-4 as Fig. 3-5. Due to space constraints, the remaining “lollipop” plots are now displayed in the Supplementary Information. Moreover, the information on how we identified and plotted these regions was included in the Methods section (p. 23, lines 805-813).

Do the authors think that PEV could be a relevant phenomenon here? Specifically, is there an opportunity to use their data to identify loci that are at the interface between faithful (active gene, demethylated), and unfaithful (silent gene, variegating methylation) states? Could there be, eg, known boundary elements between the two (mouse ENCODE has a wealth of data on this)? The authors are well-familiar with the many systems on PEV (from the classic Muller one to various subtelomeric systems, and mammalian PEV, eg Emma Whitelaw’s work among others), the classical studies on the agouti locus etc.

This is an exciting idea that we have thought about and are interested in pursuing - especially given our work on the local methylation environment surrounding metastable epialleles (<https://doi.org/10.1186/s13100-021-00235-1>). However, due to the nature of the data generated by the target capture approach, which provides deep coverage of rather sparse genomic regions (5% of the genome), we are unable to robustly identify (let alone characterise) boundaries between unfaithful and faithful methylation states. Indeed, we expect that, in the future, whole genome methylation coverage will allow us to dig deeper into the question of how, and whether, methylation states switch from being unfaithfully to faithfully inherited in a linear fashion throughout the genome.

It is unclear from the data whether there is a directionality to the unfaithfulness? In regions that are “unfaithful” – is it more likely that a methylated CpG becomes unmethylated, or the other way around? Or are both directions equally likely?

We thank the reviewer for their question highlighting a key point of the manuscript requiring textual reinforcement. The data revealed that the directionality of the unfaithfulness of methylation inheritance depends on the initial methylation level of a given unfaithful CpG. Meaning that a CpG site that is 70% methylated in a population of cells is more often likely to gain methylation in the clonally expanded lineages, while a site that is 30% methylated in the parental population of cells is more often likely to lose methylation upon clonal selection and expansion. We believe this is a key result, and therefore we have expanded and reworded a part of the discussion to clarify it (p. 11-12, lines 371-398).

//Fyodor Urnov, Innovative Genomics Institute, UC Berkeley//

Reviewer #2 (Remarks to the Author):

NCOMMS-23-10317-T Review

SUMMARY

DNA methylation, an important epigenetic mark, is found on the majority of cytosines in CpG dinucleotides in mammalian genomes, where it most often symmetrically marks the two cytosines on complementary DNA strands. Following DNA replication, the resulting hemimethylated state is recognized by and recruits the methyltransferase machinery to restore symmetric methylation and maintain the methylome between cell divisions. When assaying methylation by next-generation sequencing, using millions of cells as input, the majority of CpGs have either fully unmethylated or fully methylated readouts, indicating high homogeneity in methylation state at these loci. However, a subset of loci reliably give intermediate methylation readings, which could represent multiple possible scenarios. This is a broadly observed phenomenon, and understanding the source of intermediate methylation is of high significance, as doing so could inform on both the distribution of DNA methylation in populations and the fidelity of its propagation during mitosis.

One prevalent theory is that intermediately methylated loci represent intrapopulation heterogeneity- some cells are fully methylated and propagate this state to any and all progeny, while others are unmethylated and similarly propagate the unmethylated state. In this scenario, the methylation machinery faithfully restores all methylation to the fully symmetric state post-replication, and the intermediate methylation observed is a consequence of one population of cells being methylated and the other unmethylated.

To test this, Hay and colleagues created 2 immortalized MEF cell lines, from each of which 7 single cells were sorted and expanded to create clonal cell lines. The authors then did large-scale, targeted bisulfite sequencing (tcBS-seq) to compare methylation between parental and clonal lines. They did RNA-seq to compare methylation and expression, and additionally profiled Dnmt3a-3b DKO MEFs using tcBS-seq to assess the contribution of the de novo methyltransferases to intermediate methylation. The experimental design is elegant and straightforward, and the analyses presented are well-done and illustrated very nicely in the figures.

Based on their tcBS-seq data in clonally expanded MEFs, the authors conclude that intermediately methylated CpGs represent loci where DNA methylation is unfaithfully inherited between cell divisions. However, there are alternative explanations for the results that are not discussed. The authors repeatedly claim that their data tracks methylation inheritance between cell divisions, when their experiments do not track inheritance through mitosis or make any comparisons based on cell division/ cell cycle stage. The text should make this explicit and frame the experiments described as measuring population heterogeneity following clonal selection, which is itself an interesting and informative experimental design. To exclude the possibility that intermediate methylation is a

consequence of delayed DNA methylation maintenance following replication, the frequency and distribution of intermediate CpGs in cycling and arrested cells should also be compared (see Major Points).

We thank the reviewer for the comment on word choice to describe our experimental design. We have changed the text accordingly, replacing mentions of methylation inheritance “between cell divisions” with “through clonal expansion”, which more appropriately describes the experiment and the corresponding data.
(p1., line 17; p. 3, lines 79-81, 95; p. 5, line 173; p. 8, line 269; p. 11, line 351; p. 12, line 395, 397; p. 22, line 772; Fig. 3 title and legend; Extended Data Fig. 9 title and legend).

Major Points

Requested Experiments and Analyses

1. The authors conclude that their observations are of methylation inheritance between cell divisions, but do no analysis of cell cycle dynamics or replication timing. Given that methylation is known to be less stable/ more readily lost in late-replicating DNA, it would be informative to compare the distribution of the methylation categories defined in Figure 1C with replication timing. Replication timing domains for MEFs have already been defined (e.g., DOI: 10.1101/gr.099796.109, DOI: 10.1007/s10577-022-09703-7).

We apologise for not making this more evident in the original version of the manuscript: the target capture method we used is enriched for early replicating regions (see the response to Major Comment 4). Following the reviewer’s suggestion, we have plotted the distribution of replication timings amongst the methylation categories defined in Fig. 1C (see rebuttal Fig. A on the following page). Amongst the minority of CpGs that do reside in late-replicating regions of the genome, intermediately methylated CpGs (UI, I, and MI) are slightly more represented compared with U and M CpGs (rebuttal Fig. B). The fact that most of the tcBS-seq CpGs reside in early replicating regions of the genome indicates that although replication timing might influence methylation fidelity, it does not explain the bulk of the unfaithful methylation inheritance at intermediately methylated CpGs that we observed in our experiment.

(A) Boxplots of replication timing from publicly available MEF Repli-seq data (GSE196749) at CpGs within methylation groups previously defined in Extended Data Fig. 3A, all tcBS-seq CpGs, and all mouse genome (mm10) CpGs.
 (B) Relative distributions of replication fractions (early and late) between methylation groups.

2. In a similar vein, the Meissner lab has shown that blocking a cell population in mitosis reduces intermediate methylation (DOI: 10.1038/s41594-018-0046-4), suggesting that intermediately methylated CpGs in both parental and clonal MEF lines could represent loci that are slower to re-methylate post-DNA replication, rather than loci that differ in methylation between cells in the same cell cycle stage. To elucidate the role of DNA replication/ the cell cycle in the detection of intermediately methylated CpGs, it would be important to compare cycling and arrested populations by the tcBS-seq method used here. It would also be informative to show the cell cycle distribution of the MEF-1 and MEF-2 cell lines.

As we have shown from the analysis performed in response to the reviewer's previous comment, the CpGs probed by the tcBS-seq method predominantly reside in early-replicating regions of the genome, suggesting a minimal role of DNA replication timing in understanding intermediate methylation states from tcBS-seq data. Nevertheless, we think this is an experiment worth pursuing for a future study that considers the whole genome without specific biases towards early-replicating regions.

3. Did anything distinguish the CpGs that stably differed in methylation status between MEF-1 and MEF-2? And, did anything distinguish the CpGs that differed between parental and clonal lines? How many such CpGs were identified?

This is an intriguing question. When testing for genomic features or transcriptional profiles, we could not find anything distinguishing CpGs that stably differ between MEF-1 and MEF-2. As for CpGs that differ between parental and clonal lines, the majority of faithfully hypomethylated and hypermethylated CpGs (~300,000 and ~200,000 respectively) tend to show high similarity between parents and clones (Fig. 3C [now Fig 4C]; low and high). On the other hand, it is a very rare occurrence for intermediately methylated CpGs (~3000) to have consistently similar methylation levels between the parental lines and all the clonal lines, while for the vast majority (~300,000), at least one clonal line is different from the parental (Fig. 3C [now Fig 4C]; low intermediate, intermediate, and high intermediate).

4. The authors use an enrichment-based method, tcBS-seq, rather than an unbiased whole-genome approach. Can some characterization of the probes used (how representative they are of the genome regarding CpG density, genic/intergenic/TE proportions etc) be included, to better contextualize the loci studied here?

We apologise for not initially providing a more thorough characterisation of the regions targeted by our tcBS-seq probes. We have added Extended Data Table 1 (renumbering existing Extended Data Tables from 1-4 to 2-5) that shows tcBS-seq coverage by CpGs across different annotations and functional categories (genic, transposable element, CpG island, replication timing). As previously mentioned, the tcBS-seq CpGs are enriched in genic and early-replicating regions and are underrepresented at transposable elements – we have added a sentence to explain this in the text (see p. 3, lines 89-91). We think the new table complements Extended Data Table 2 (formerly Extended Data Table 1), contextualizing the tcBS-seq coverage by region (e.g. how many promoters are covered by at least 1 CpG).

Requested Edits in Text

5. As depicted in Figure 1, the authors only consider symmetric methylation states in their interpretation of the data. However, the tcBS-seq method wouldn't be able to discriminate monoallelic methylation from biallelic hemi-methylation, and a single cell could also conceivably have 25% or 75% methylation at a single locus if one allele is asymmetrically methylated and the other is symmetrically methylated/ unmethylated. Whether or not hemi-methylated CpGs in any of these contexts (originating from DNA replication, or de novo methylation gain/loss on a single DNA strand) could underlie intermediate methylation at a population level should be discussed.

This is a great point, and as suggested by the reviewer, we have included a few sentences discussing the potential role that hemimethylation plays in the unfaithful inheritance of intermediately methylated CpGs (see p. 11, lines 359-366).

6. In this work, population heterogeneity following clonal selection is analyzed to gain insights into the faithfulness of methylation propagation in cycling cells. However, several groups have now directly analyzed methylation on replicated DNA over time by sequencing or mass spectrometry. These experiments assess mitotic inheritance of DNA methylation, and should be included in the cited literature to contextualize the findings of this study.

Thank you for pointing out these omissions. We have added these references as a part of the new paragraph in the discussion that also relates to the previous comment regarding hemimethylation (see p. 11, lines 363-366).

7. Can the authors comment on the selection of MEFs for this study, given that more arguably homogeneous cell types (e.g., mESCs) also have well-documented regions of intermediate CpG methylation?

We used freshly derived immortalised MEFs as a model to study methylation inheritance of intermediately methylated sites in the genome due to their capacity to replicate indefinitely, as well as their biological proximity to primary tissues and primary cell lines compared with cancer cells or commercially available lines. Furthermore, MEFs come from a developmental context in which global methylation levels are relatively stabilised (E13.5). On the other hand, mESCs (for example) are derived from blastocysts where methylation levels are in flux making them a complex model to reliably study DNA methylation population heterogeneity and inheritance through clonal expansion.

8. The Methods and Extended Data Tables 2 and 3 refer to cultured mESC/ E3.5 data that isn't presented in any of the figures, so these references should be removed.

Thank you for bringing this to our attention - we have removed these references.

9. Regarding Extended Data 10, it is important to point out that though H3K27me3 and H3K9me3 are both associated with repression, H3K27me3 tends to anticorrelate with DNA methylation and H3K9me3 positively correlates with it. The H3K9me3 association with intermediate methylation is interesting, given that direct quantification of DNA methylation on DNA pulled down with H3K9me3-marked nucleosomes shows very high methylation levels (DOI: 10.1038/s41556-022-01048-x). Could these be intermediate CpGs as a consequence of their residing in late-replicating regions?

As explained in our responses to comments 1, 2, and 4, the CpGs analysed in our study are overwhelmingly enriched for early-replicating regions of the genome.

Minor Points

10. Is the text describing relative genome coverage correct in the Extended Data Fig 1 legend? Based on the figure MEF-1 (red) has increased coverage at chr12/18/x and decreased coverage at chr19, but this is flipped in the text.

The text describing relative genome coverage in Extended Data Fig. 1 is indeed correct. However, given the reviewer's comment, we provide further explanation in the Methods section on p. 20 of the revised manuscript, lines 722-725.

First, we calculated the mean coverage of a CpG as the average coverage across parental and clonal cell lines. Next, we normalised the data by dividing the mean coverage of each CpG by the sum of all mean coverages (e.g. for MEF-1, this sum is equal to 50,975,536), which yields a very small number. For example, if the mean coverage of a CpG is 40, the normalised mean coverage would be $40 / 50,975,536 \cong 7.85e-7$; the $-\log_2(7.85e-7) \cong 20.3$. Meanwhile, a CpG with a mean coverage of 20 would have a normalised mean coverage of $3.92e-7$, with a $-\log_2 \cong 21.3$. This is all to explain that lower values of $-\log_2(\text{normalised mean coverage of a CpG})$ represent higher coverage and vice-versa.

Furthermore, summing all normalised mean coverages should yield a value of 1. In the original Extended Data Fig. 1A, we made the mistake of normalising it to 0.125. We have now corrected Extended Data Fig. 1 to show $-\log_2$ values that reflect a normalisation of mean coverage to 1 instead.

11. In Figures 1G and 1H, it seems that individual CpGs can both have an expression designation and be intergenic. Since the expression is over protein-coding genes, shouldn't these be mutually exclusive?

As the reviewer suggests, the CpGs that are highlighted in the expression analysis (Fig. 1G [now Fig. 1D]) and those that are highlighted as intergenic (Fig. 1H [now Fig. 1D]) are indeed mutually exclusive. Because there are >1.2 million rows (each one representing a CpG), we are limited by the resolution of the heatmap. Therefore, it may look like there are shared CpGs between these two columns in Fig 1G and 1H (now Fig. 1D), when in reality they are not.

12. Extended Data Tables 1 and 2 don't seem to relate to the text in which they're referenced in the Results section.

Thank you for bringing this to our attention - we have removed these references.

13. In Extended Data 2E, could the difference in 3T3 MEFs compared to the other datasets be briefly commented on?

3T3 MEFs were established in the 1960s, and the methylation data have been previously published (GSM4942823, GSM4942824, GSM4942825). They appear to have a markedly different global methylation profile from both our immortalised MEFs, primary MEFs, and E13.5/E14.5 mouse limb (Extended Data Fig. 2 A-D) - including at SINEs (Extended Data Fig. 2E), as the reviewer points out. Using imprinting control regions as a barometer for epigenetic health, we found that these 3T3 cells, unlike any of the other tested datasets, are hypermethylated as opposed to intermediately methylated, which in this case represents an allelic bias (Extended Data Fig. 2C).

14. In the Main Text, paragraph 3, please cite the panels being discussed in the second half of the paragraph for clarity.

As the reviewer has suggested, we have added a reference to Fig. 1C to paragraph 3 of the main text (see p. 4, line 151).

15. In Extended Data 3C-E, could the datasets being compared statistically please be depicted more clearly in the figures?

We apologise for the confusion and have depicted which datasets are being compared statistically more clearly in Extended Data Figures 3C-E, modifying the legend accordingly.

16. In Extended Data 4A, there's a mix of brackets and parentheses that should be standardized.

In mathematical interval notation for datasets, square brackets indicate that the boundary data point is included within the indicated dataset (inclusive), while the parentheses indicate that the boundary data point is not included (exclusive). We have added this information to the legend of Extended Data Fig. 4.

17. In Figure 2A/B and Extended Data 6A/B, is the interquartile range only shown for the “none” and “high” expression categories? This is how it appears in the graphs, but should be made clear if so in the legends.

In Fig. 2A/B (now Fig. 3A/B) and Extended Data Fig. 6A/B, the interquartile range is indeed only shown for the “none” and “high” gene expression categories. As the reviewer has suggested, we have made this clear in the figure legends (see p. 15, lines 607-608 and the Extended Data).

18. The text referencing Extended Data 8 indicates that the figure depicts the proportion of TEs in exons, but it does not.

Thank you for pointing this out - we have moved the reference to Extended Data Fig. 8 to a more appropriate place in the text (see p. 7 line 252).

19. The key in Figure 3B should be copied into Extended Data Figure 9A to help with interpretation.

We have added the key in Fig. 3B (now Fig. 4B) to Extended Data Fig. 9A and modified Extended Data Figure 9A and B to have the same formatting as the corresponding Fig. 3B and C (now Fig. 4B and C).

20. In the legends for Figure 3C and Extended Data 9B, it should be stated what the green dots and lines in the UpSet plot represent, just for completeness.

Thank you for pointing this out. We have modified the legends of both Fig. 3 (now Fig. 4) and Extended Data Fig. 9 to state that the green dots and lines in the UpSet plots represent cases and combinations of potential faithful methylation inheritance (p. 16, lines 631-632 and Extended Data).

21. The discussion of statistics and trends in the DKO data could be more clear as a figure, rather than being solely in the text.

Because we did not find a meaningful difference in methylation between the DKO and control datasets, we opted not to include a figure representing the statistics discussed in the text.

REVIEWERS' COMMENTS

Reviewer #1 (Remarks to the Author):

The revised manuscript fully addresses comments raised during initial review. The new Fig 2 is a striking representation of the phenomenon the authors discovered and characterized.

//fdu//

Reviewer #2 (Remarks to the Author):

The authors have adequately addressed all comments, and I am happy to recommend this manuscript for publication.